# Nonconvex Low-Rank Symmetric Tensor Completion from Noisy Data

**Changxiao Cai**
Princeton University

**Gen Li**
Tsinghua University

**H. Vincent Poor**
Princeton University

**Yuxin Chen**
Princeton University

## Abstract

We study a completion problem of broad practical interest: the reconstruction of a low-rank symmetric tensor from highly incomplete and randomly corrupted observations of its entries. While a variety of prior work has been dedicated to this problem, prior algorithms either are computationally too expensive for large-scale applications, or come with sub-optimal statistical guarantees. Focusing on "incoherent" and well-conditioned tensors of a constant CP rank, we propose a two-stage nonconvex algorithm — (vanilla) gradient descent following a rough initialization — that achieves the best of both worlds. Specifically, the proposed nonconvex algorithm faithfully completes the tensor and retrieves individual tensor factors within nearly linear time, while at the same time enjoying near-optimal statistical guarantees (i.e. minimal sample complexity and optimal $\ell_2$ and $\ell_\infty$ statistical accuracy). The insights conveyed through our analysis of nonconvex optimization might have implications for other tensor estimation problems.

## 1 Introduction

### 1.1 Tensor completion from noisy entries

Estimation of low-complexity models from highly incomplete observations is a fundamental task that spans a diverse array of engineering applications. Arguably one of the most extensively studied problems of this kind is matrix completion, where one wishes to recover a low-rank matrix given only partial entries [21, 14]. Moving beyond matrix-type data, a natural higher-order generalization is *low-rank tensor completion*, which aims to reconstruct a low-rank tensor when the vast majority of its entries are unseen. There is certainly no shortage of applications that motivate the investigation of tensor completion, examples including seismic data analysis [44, 24], visual data in-painting [47, 46], medical imaging [25, 58, 19], multi-dimensional harmonic retrieval [13, 72], to name just a few.

For the sake of clarity, we phrase the problem formally before we proceed, focusing on a simple model that already captures the intrinsic difficulty of tensor completion in many aspects.[1] Imagine we are asked to estimate a symmetric order-three tensor[2] $\boldsymbol{T}^\star \in \mathbb{R}^{d \times d \times d}$ from a few noisy entries

$$T_{j,k,l} = T_{j,k,l}^\star + E_{j,k,l}, \qquad \forall (j, k, l) \in \Omega, \tag{1}$$

where $T_{j,k,l}$ is the observed noisy entry at location $(j, k, l)$, $E_{j,k,l}$ stands for the associated noise, and $\Omega \subseteq \{1, \cdots, d\}^3$ is a symmetric index subset to sample from. For notational simplicity, we set $\boldsymbol{T} = [T_{j,k,l}]_{1 \leq j,k,l \leq d}$ and $\boldsymbol{E} = [E_{j,k,l}]_{1 \leq j,k,l \leq d}$, with $T_{j,k,l} = E_{j,k,l} = 0$ for any $(j, k, l) \notin \Omega$. We adopt a *random sampling* model such that each index $(j, k, l)$ $(j \leq k \leq l)$ is included in $\Omega$ independently with probability $p$. In addition, we know *a priori* that the unknown tensor $\boldsymbol{T}^\star \in \mathbb{R}^{d \times d \times d}$ is a superposition of $r$ rank-one tensors (often termed canonical polyadic (CP) decomposition if $r$ is minimal)

$$\boldsymbol{T}^\star = \sum\nolimits_{i=1}^{r} \boldsymbol{u}_i^\star \otimes \boldsymbol{u}_i^\star \otimes \boldsymbol{u}_i^\star, \qquad \text{or more concisely,} \qquad \boldsymbol{T}^\star = \sum\nolimits_{i=1}^{r} \boldsymbol{u}_i^{\star \otimes 3}, \tag{2}$$

where each $\boldsymbol{u}_i^\star \in \mathbb{R}^d$ represents one of the $r$ factors. The primary question is: can we hope to faithfully estimate $\boldsymbol{T}^\star$, as well as the factors $\{\boldsymbol{u}_i^\star\}_{1\leq i\leq r}$, from the partially revealed entries (1)?

## 1.2 Computational and statistical challenges

Even though tensor completion conceptually resembles matrix completion in various ways, it is considerably more challenging than the matrix counterpart. This is perhaps not surprising, given that a plethora of natural tensor problems are all notoriously hard [32]. As a notable example, while matrix completion is often efficiently solvable under nearly minimal sample complexity [8, 29], all polynomial-time algorithms developed so far for tensor completion — even in the noise-free case — require a sample size at least exceeding the order of $rd^{3/2}$. This is substantially larger than the degrees of freedom (i.e. $rd$) underlying the model (2). In fact, it is widely conjectured that there exists a large computational barrier away from the information-theoretic sampling limits [4].

With this fundamental gap in mind, the current paper focuses on the regime (in terms of the sample size) that enables reliable tensor completion in polynomial time. A variety of algorithms have been proposed that enjoy some sort of theoretical guarantees in (at least part of) this regime, including but not limited to spectral methods [50], sum-of-squares hierarchy [4, 53], nonconvex algorithms [36, 67], and also convex relaxation (based on proper unfolding) [25, 64, 34, 57, 47, 51, 28]. While these are all polynomial-time algorithms, most of the computational complexities supported by prior theory remain prohibitively high when dealing with large-scale tensor data. The only exception is the unfolding-based spectral method, which, however, fails to achieve exact recovery even when the noise vanishes. This leads to a critical question that this paper aims to explore:

> **Q1:** *Is there any linear-time algorithm that is guaranteed to work for tensor completion?*

Going beyond such computational concerns, one might naturally wonder whether it is also possible for a fast algorithm to achieve a nearly un-improvable statistical accuracy in the presence of noise. Towards this end, intriguing stability guarantees have been established for sum-of-squares hierarchy in the noisy settings [4], although this paradigm is computationally prohibitive for large-scale data. The recent work [68] came up with a two-stage algorithm (i.e. spectral method followed by tensor power iterations) for noisy tensor completion. Its estimation accuracy, however, falls short of achieving exact recovery in the absence of noise. This gives rise to another question of fundamental importance:

> **Q2:** *Can we achieve near-optimal statistical accuracy without compromising computational efficiency?*

## 1.3 A two-stage nonconvex algorithm

To address the above-mentioned challenges, a first impulse is to resort to the least squares formulation

$$\underset{\boldsymbol{u}_1,\cdots,\boldsymbol{u}_r\in\mathbb{R}^d}{\text{minimize}} \quad \sum_{j,k,l\in\Omega} \left( \left[\sum_{i=1}^r \boldsymbol{u}_i^{\otimes 3}\right]_{j,k,l} - T_{j,k,l} \right)^2, \tag{3}$$

or more concisely (up to proper re-scaling),

$$\underset{\boldsymbol{U}\in\mathbb{R}^{d\times r}}{\text{minimize}} \quad f(\boldsymbol{U}) \triangleq \frac{1}{6p} \left\| \mathcal{P}_\Omega\left( \sum_{i=1}^r \boldsymbol{u}_i^{\otimes 3} - \boldsymbol{T} \right) \right\|_{\mathrm{F}}^2 \tag{4}$$

if we take $\boldsymbol{U} := [\boldsymbol{u}_1,\ldots,\boldsymbol{u}_r] \in \mathbb{R}^{d\times r}$. Here, we denote by $\mathcal{P}_\Omega(\boldsymbol{T})$ the orthogonal projection of any tensor $\boldsymbol{T}$ onto the subspace of tensors which vanish outside of $\Omega$. This optimization problem, however, is highly nonconvex, resulting in computational intractability in general.

Fortunately, not all nonconvex problems are as danuting as they may seem. For example, recent years have seen a flurry of activity in low-rank matrix factorization via nonconvex optimization, which achieves optimal statistical and computational efficiency at once [55, 39, 41, 35, 9, 12, 62, 20, 18, 11, 76, 49, 65, 78]. Motivated by this strand of work, we propose to solve (4) via a two-stage nonconvex paradigm, presented below in reverse order. The procedure is summarized in Algorithms 1-3.

**Gradient descent (GD).** Arguably one of the simplest optimization algorithms is gradient descent, which adopts a gradient update rule

$$\boldsymbol{U}^{t+1} = \boldsymbol{U}^t - \eta_t \nabla f(\boldsymbol{U}^t), \qquad t = 0, 1, \cdots \tag{5}$$

---

**Algorithm 1** Gradient descent for nonconvex tensor completion

1: **Input:** observed entries $\{T_{j,k,l} \mid (j,k,l) \in \Omega\}$, sampling rate $p$, number of iterations $t_0$.
2: Generate an initial estimate $\boldsymbol{U}^0 \in \mathbb{R}^{d \times r}$ via Algorithm 2.
3: **for** $t = 0, 1, \ldots, t_0 - 1$ **do**
4: $\qquad \boldsymbol{U}^{t+1} = \boldsymbol{U}^t - \eta_t \nabla f(\boldsymbol{U}^t) = \boldsymbol{U}^t - \frac{\eta_t}{p} \mathcal{P}_\Omega \big( \sum_{i=1}^{r} (\boldsymbol{u}_i^t)^{\otimes 3} - \boldsymbol{T} \big) \times_1^{\mathsf{seq}} \boldsymbol{U}^t \times_2^{\mathsf{seq}} \boldsymbol{U}^t$, where $\times_1^{\mathsf{seq}}$
$\qquad$ and $\times_2^{\mathsf{seq}}$ are defined in Section 1.5.

---

**Algorithm 2** Spectral initialization for nonconvex tensor completion

1: **Input:** sampling set $\Omega$, observed entries $\{T_{i,j,k} \mid (i,j,k) \in \Omega\}$, sampling rate $p$.
2: Let $\boldsymbol{U\Lambda U}^\top$ be the rank-$r$ eigen-decomposition of

$$\boldsymbol{B} := \mathcal{P}_{\mathsf{off\text{-}diag}}(\boldsymbol{AA}^\top), \qquad\qquad (6)$$

$\qquad$ where $\boldsymbol{A} = \mathsf{unfold}\big(p^{-1}\boldsymbol{T}\big)$ is the mode-1 matricization of $p^{-1}\boldsymbol{T}$, and $\mathcal{P}_{\mathsf{off\text{-}diag}}(\boldsymbol{Z})$ extracts out
$\qquad$ the off-diagonal entries of $\boldsymbol{Z}$.
3: **Output:** initial estimate $\boldsymbol{U}^0 \in \mathbb{R}^{d \times r}$ from $\boldsymbol{U} \in \mathbb{R}^{d \times r}$ using Algorithm 3.

---

where $\eta_t$ is the learning rate. The main computational burden in each iteration lies in gradient evaluation, which, in this case, can be performed in time proportional to that taken to read the data.

Despite the simplicity of this algorithm, two critical issues stand out and might significantly affect its efficiency, which we shall bear in mind throughout the algorithmic and theoretical development.

*(i) Local stationary points and initialization.* As is well known, GD is guaranteed to find an approximate local stationary point, provided that the learning rates do not exceed the inverse Lipschitz constant of the gradient [5]. There exist, however, local stationary points (e.g. saddle points or spurious local minima) that might fall short of the desired statistical properties. This requires us to properly avoid such undesired points, while retaining computational efficiency. To address this issue, one strategy is to first identify a rough initial guess within a local region surrounding the global solution, which often helps rule out bad local minima. As a side remark, while careful initialization might not be crucial for several matrix recovery cases [45, 15, 27], it does seem to be critical in various tensor problems [56]. We shall elucidate this point in the full version [7].

*(ii) Learning rates and regularization.* Learning rates play a pivotal role in determining the convergence properties of GD. The challenge, however, is that the loss function (4) is overall not sufficiently smooth (i.e. its gradient often has a very large Lipschitz constant), and hence generic optimization theory recommends a pessimistically slow update rule (i.e. an extremely small learning rate) so as to guard against over-shooting. This, however, slows down the algorithm significantly, thus destroying the main computational advantage of GD (i.e. low per-iteration cost). With this issue in mind, prior literature suggests carefully designed regularization steps (e.g. proper projection, regularized loss functions) in order to improve the geometry of the optimization landscape [67]. By contrast, we argue that one is allowed to take a constant learning rate — which is as aggressive as it can possibly be — even without enforcing any regularization procedures.

**Initialization.** Motivated by the above-mentioned issue (i), we develop a procedure that guarantees a reasonable initial estimate. In a nutshell, the proposed procedure consists of two steps:

(a) Estimate the subspace spanned by the $r$ tensor factors $\{\boldsymbol{u}_i^\star\}_{1 \leq i \leq r}$ via a spectral method;

(b) Disentangle individual tensor factors from this subspace estimate.

The computational complexity of the proposed initialization is linear-time (i.e. $O(pd^3)$) when $r = O(1)$. Note, however, that these steps are more complicated to describe. We postpone the details to Section 2 and intuitions to [7]. The readers can catch a glimpse of these procedures in Algorithm 2-3.

## 1.4 Main results

Encouragingly, the proposed nonconvex algorithm provably achieves the best of both worlds — in terms of statistical accuracy and computational efficiency — for a broad class of problem instances.

---
**Algorithm 3** Retrieval of low-rank tensor factors from a given subspace estimate.

---
1: **Input:** sampling set $\Omega$, observed entries $\{T_{i,j,k} \mid (i,j,k) \in \Omega\}$, sampling rate $p$, number of restarts $L$, pruning threshold $\epsilon_{\mathsf{th}}$, subspace estimate $\boldsymbol{U} \in \mathbb{R}^{d \times r}$.
2: **for** $\tau = 1, \ldots, L$ **do**
3:      Generate an independent Gaussian vector $\boldsymbol{g}^\tau \sim \mathcal{N}(0, \boldsymbol{I}_d)$.
4:      $\left(\boldsymbol{\nu}^\tau, \lambda_\tau, \mathsf{spec\text{-}gap}_\tau\right) \leftarrow$ RETRIEVE-ONE-TENSOR-FACTOR$(\boldsymbol{T}, p, \boldsymbol{U}, \boldsymbol{g}^\tau)$.
5: Generate $\left\{(\boldsymbol{w}^1, \lambda_1), \ldots, (\boldsymbol{w}^r, \lambda_r)\right\} \leftarrow$ PRUNE$\left(\left\{\left(\boldsymbol{\nu}^\tau, \lambda_\tau, \mathsf{spec\text{-}gap}_\tau\right)\right\}_{\tau=1}^L, \epsilon_{\mathsf{th}}\right)$.
6: **Output:** initial estimate $\boldsymbol{U}^0 = \left[\lambda_1^{1/3} \boldsymbol{w}^1, \ldots, \lambda_r^{1/3} \boldsymbol{w}^r\right]$.

---

---
1: **function** RETRIEVE-ONE-TENSOR-FACTOR$(\boldsymbol{T}, p, \boldsymbol{U}, \boldsymbol{g})$
     Compute

$$\boldsymbol{\theta} = \boldsymbol{U}\boldsymbol{U}^\top \boldsymbol{g} =: \mathcal{P}_{\boldsymbol{U}}(\boldsymbol{g}), \tag{7a}$$

$$\boldsymbol{M} = p^{-1}\boldsymbol{T} \times_3 \boldsymbol{\theta}, \tag{7b}$$

where $\times_3$ is defined in Section 1.5.
2:      Let $\boldsymbol{\nu}$ be the leading singular vector of $\boldsymbol{M}$ obeying $\langle \boldsymbol{T}, \boldsymbol{\nu}^{\otimes 3}\rangle \geq 0$; set $\lambda = \langle p^{-1}\boldsymbol{T}, \boldsymbol{\nu}^{\otimes 3}\rangle$.
3:      **return** $\left(\boldsymbol{\nu}, \lambda, \sigma_1(\boldsymbol{M}) - \sigma_2(\boldsymbol{M})\right)$.

---

Before continuing, we note that one cannot hope to recover an arbitrary tensor from highly sub-sampled and arbitrarily corrupted entries. In order to enable provably valid recovery, the present paper focuses on a tractable model by imposing the following assumptions.

**Assumption 1.1** (Incoherence and well-conditionedness). *The tensor factors $\{\boldsymbol{u}_i^\star\}_{1 \leq i \leq r}$ satisfy*

$$(\mathbf{A1}) \quad \|\boldsymbol{T}^\star\|_\infty \leq \sqrt{\mu_0/d^3}\, \|\boldsymbol{T}^\star\|_{\mathrm{F}}, \tag{8a}$$

$$(\mathbf{A2}) \quad \|\boldsymbol{u}_i^\star\|_\infty \leq \sqrt{\mu_1/d}\, \|\boldsymbol{u}_i^\star\|_2, \qquad\qquad 1 \leq i \leq d; \tag{8b}$$

$$(\mathbf{A3}) \quad |\langle \boldsymbol{u}_i^\star, \boldsymbol{u}_j^\star\rangle| \leq \sqrt{\mu_2/d}\, \|\boldsymbol{u}_i^\star\|_2 \|\boldsymbol{u}_j^\star\|_2, \qquad 1 \leq i \neq j \leq d; \tag{8c}$$

$$(\mathbf{A4}) \quad \kappa \triangleq \left(\max_i \|\boldsymbol{u}_i^\star\|_2^3\right) / \left(\min_i \|\boldsymbol{u}_i^\star\|_2^3\right) = O(1). \tag{8d}$$

**Remark 1.2.** *Here, $\mu_0, \mu_1$ and $\mu_2$ are termed the incoherence parameters. Assumptions $\mathbf{A1}$, $\mathbf{A2}$ and $\mathbf{A3}$ can be viewed as some sort of incoherence conditions for the tensor. For instance, when $\mu_0, \mu_1$ and $\mu_2$ are small, these conditions say that (1) the energy of tensor $\boldsymbol{T}^\star$ is (nearly) evenly spread across all entries; (2) each factor $\boldsymbol{u}_i^\star$ is de-localized; (3) the factors $\{\boldsymbol{u}_i^\star\}$ are nearly orthogonal to each other. Assumption $\mathbf{A4}$ is concerned with the "well-conditionedness" of the tensor, meaning that each rank-1 component is of roughly the same strength.*

For notational simplicity, we shall set $\mu := \max\{\mu_0, \mu_1, \mu_2\}$.

**Assumption 1.3** (Random noise). *Suppose that $\boldsymbol{E}$ is a symmetric random tensor, where $\{E_{j,k,l}\}_{1 \leq j \leq k \leq l \leq d}$ (cf. (1)) are independently generated symmetric sub-Gaussian random variables with mean zero and variance $\mathsf{Var}(E_{j,k,l}) \leq \sigma^2$.*

In addition, recognizing that there is a global permutational ambiguity issue (namely, one cannot distinguish $\boldsymbol{u}_1^\star, \cdots, \boldsymbol{u}_r^\star$ from an arbitrary permutation of them), we introduce the following loss metrics to account for this ambiguity

$$\mathsf{dist}_{\mathrm{F}}(\boldsymbol{U}, \boldsymbol{U}^\star) := \min_{\boldsymbol{\Pi} \in \mathsf{perm}_r} \|\boldsymbol{U}\boldsymbol{\Pi} - \boldsymbol{U}^\star\|_{\mathrm{F}}, \tag{9a}$$

---
1: **function** PRUNE$\left(\left\{\left(\boldsymbol{\nu}^\tau, \lambda_\tau, \mathsf{spec\text{-}gap}_\tau\right)\right\}_{\tau=1}^L, \epsilon_{\mathsf{th}}\right)$
2:      Set $\Theta = \left\{\left(\boldsymbol{\nu}^\tau, \lambda_\tau, \mathsf{spec\text{-}gap}_\tau\right)\right\}_{\tau=1}^L$.
3:      **for** $i = 1, \ldots, r$ **do**
4:          Choose $(\boldsymbol{\nu}^\tau, \lambda_\tau, \mathsf{spec\text{-}gap}_\tau)$ from $\Theta$ with the largest $\mathsf{spec\text{-}gap}_\tau$; set $\boldsymbol{w}^i = \boldsymbol{\nu}^\tau, \lambda_i = \lambda_\tau$.
5:          Update $\Theta \leftarrow \Theta \setminus \left\{\left(\boldsymbol{\nu}^\tau, \lambda_\tau, \mathsf{spec\text{-}gap}_\tau\right) \in \Theta : |\langle \boldsymbol{\nu}^\tau, \boldsymbol{w}^i\rangle| > 1 - \epsilon_{\mathsf{th}}\right\}$.
6:      **return** $\left\{(\boldsymbol{w}^1, \lambda_1), \ldots, (\boldsymbol{w}^r, \lambda_r)\right\}$.

---

$$\mathsf{dist}_\infty(\boldsymbol{U}, \boldsymbol{U}^\star) := \min_{\boldsymbol{\Pi} \in \mathsf{perm}_r} \|\boldsymbol{U}\boldsymbol{\Pi} - \boldsymbol{U}^\star\|_\infty, \tag{9b}$$

$$\mathsf{dist}_{2,\infty}(\boldsymbol{U}, \boldsymbol{U}^\star) := \min_{\boldsymbol{\Pi} \in \mathsf{perm}_r} \|\boldsymbol{U}\boldsymbol{\Pi} - \boldsymbol{U}^\star\|_{2,\infty}, \tag{9c}$$

where $\mathsf{perm}_r$ stands for the set of $r \times r$ permutation matrices. For notational simplicity, we also take $\lambda_{\min}^\star := \min_{1 \le i \le r} \|\boldsymbol{u}_i^\star\|_2^3$ and $\lambda_{\max}^\star := \max_{1 \le i \le r} \|\boldsymbol{u}_i^\star\|_2^3$.

With these in place, we are ready to present our main results.

**Theorem 1.4.** *Fix an arbitrary small constant $\delta > 0$. Suppose that $r, \kappa, \mu = O(1)$,*

$$p \ge c_0 \frac{\log^5 d}{d^{3/2}}, \qquad \frac{\sigma}{\lambda_{\min}^\star} \le c_1 \frac{\sqrt{p}}{\sqrt{d^{3/2} \log^5 d}},$$

$$L = c_3 \quad and \quad \epsilon_{\mathsf{th}} = c_4 \left( \frac{\log d}{d\sqrt{p}} + \frac{\sigma}{\lambda_{\min}^\star} \sqrt{\frac{d \log^2 d}{p}} + \sqrt{\frac{\log d}{d}} \right)$$

*for some sufficiently large constants $c_0, c_3 > 0$ and some sufficiently small constants $c_1, c_4 > 0$. The learning rate $\eta_t \equiv \eta$ is taken to be a constant obeying $0 < \eta \le \lambda_{\min}^{\star 4/3} / (32\lambda_{\max}^{\star 8/3})$. Then with probability at least $1 - \delta$,*

$$\mathsf{dist}_{\mathrm{F}}(\boldsymbol{U}^t, \boldsymbol{U}^\star) \le \left( C_1 \rho^t + C_2 \frac{\sigma}{\lambda_{\min}^\star} \sqrt{\frac{d \log d}{p}} \right) \|\boldsymbol{U}^\star\|_{\mathrm{F}} \tag{10a}$$

$$\mathsf{dist}_\infty(\boldsymbol{U}^t, \boldsymbol{U}^\star) \le \mathsf{dist}_{2,\infty}(\boldsymbol{U}^t, \boldsymbol{U}^\star) \le \left( C_3 \rho^t + C_4 \frac{\sigma}{\lambda_{\min}^\star} \sqrt{\frac{d \log d}{p}} \right) \|\boldsymbol{U}^\star\|_{2,\infty} \tag{10b}$$

*hold simultaneously for all $0 \le t \le t_0 = d^5$. Here, $0 < C_1, C_3, \rho < 1$ and $C_2, C_4 > 0$ are some absolute constants.*

*Proof.* The proof of this theorem is built upon a powerful statistical technique — called the leave-one-out analysis [23, 16, 1, 49, 79, 15, 22, 17, 52]. The proof can be found in our full version [7]. □

Several important implications are as follows. The discussion below assumes $\lambda_{\max}^\star \asymp \lambda_{\min}^\star \asymp 1$ for notational simplicity.

- *Linear convergence.* In the absence of noise, the proposed algorithm converges linearly, namely, it provably attains $\varepsilon$ accuracy within $O(\log(1/\varepsilon))$ iterations. Given the inexpensiveness of each gradient iteration, this algorithm can be viewed as a linear-time algorithm, which can almost be implemented as long as we can read the data.

- *Near-optimal sample complexity.* The fast convergence is guaranteed as soon as the sample size exceeds the order of $d^{3/2}\mathrm{poly}\log(d)$. This matches the minimal sample complexity — modulo some logarithmic factor — known so far for any polynomial-time algorithm.

- *Near-optimal statistical accuracy.* The proposed algorithm converges geometrically fast to a point with Euclidean error $O(\sigma\sqrt{(d\log d)/p})$. This matches the lower bound established in [68, Theorem 5] up to some logarithmic factor.

- *Entrywise estimation accuracy.* In addition to the Euclidean error bound, we have also established an entrywise error bound which, to the best of our knowledge, has not been established in any of the prior works. When $t$ is sufficiently large, the iterates reach an entrywise error bound $O(\sigma\sqrt{(\log d)/p})$. This entrywise error bound is about $\sqrt{d}$ times smaller than the above $\ell_2$ norm bound, implying that the estimation errors are evenly spread out across all entries.

- *Implicit regularization.* One appealing feature of our finding is the simplicity of the algorithm. All of the above statistical and computational benefits hold for vanilla gradient descent (when properly initialized). This should be contrasted with prior work (e.g. [67]) that requires extra regularization to stabilize the optimization landscape. In principle, vanilla GD implicitly constrains itself within a region of well-conditioned landscape, thus enabling fast convergence without regularization.

- *No sample splitting.* The theory developed herein does not require fresh samples in each iteration. We note that sample splitting has been frequently adopted in other context primarily to simplify analysis. Nevertheless, it typically does not exploit the data in an efficient manner (i.e. each data sample is used only once), thus resulting in the need of a much larger sample size in practice.

As an immediate consequence of Theorem 1.4, we obtain optimal $\ell_\infty$ statistical guarantees for estimating tensor entries, which are previously rarely available (see Table 1). Specifically, let our tensor estimate in the $t$-th iteration be $\boldsymbol{T}^t := \sum_{i=1}^r \boldsymbol{u}_i^t \otimes \boldsymbol{u}_i^t \otimes \boldsymbol{u}_i^t$, where $\boldsymbol{U}^t = [\boldsymbol{u}_1^t, \cdots, \boldsymbol{u}_r^t] \in \mathbb{R}^{d \times r}$.

| | algorithm | sample complexity | comput. complexity | $\ell_2$ error (noisy) | $\ell_\infty$ error (noisy) | recovery type (noiseless) |
|---|---|---|---|---|---|---|
| ours | spectral method + (vanilla) GD | $d^{1.5}$ | $pd^3$ | $\sigma\sqrt{\frac{d}{p}}$ | $\sigma\sqrt{\frac{1}{p}}$ | exact |
| [68] | spectral initialization + tensor power method | $d^{1.5}$ | $pd^3$ | $\frac{(\|\boldsymbol{T}^\star\|_\infty+\sigma)\sqrt{d}}{\sqrt{p}}$ | n/a | approximate |
| [67] | spectral method + GD on manifold | $d^{1.5}$ | $\mathrm{poly}(d)$ | n/a | n/a | exact |
| [50] | spectral method | $d^{1.5}$ | $d^3$ | n/a | n/a | approximate |
| [4] | sum-of-squares | $d^{1.5}$ | $d^{15}$ | $\frac{\|\boldsymbol{T}^\star\|_{\mathrm{F}}}{\sqrt{pd^{1.5}}} + \sigma d^{1.5}$ | n/a | approximate |
| [53] | sum-of-squares | $d^{1.5}$ | $d^{10}$ | n/a | n/a | exact |
| [73] [74] | tensor nuclear norm minimization | $d$ | NP-hard | n/a | n/a | exact |

Table 1: Comparison with theory for existing methods when $r, \mu, \kappa \asymp 1$ (neglecting log factors).

**Corollary 1.5.** *Fix an arbitrarily small constant $\delta > 0$. Instate the assumptions of Theorem 1.4. Then with probability at least $1 - \delta$,*

$$\left\|\boldsymbol{T}^t - \boldsymbol{T}^\star\right\|_{\mathrm{F}} \lesssim \left(C_1\rho^t + C_2\frac{\sigma}{\lambda_{\min}^\star}\sqrt{\frac{d\log d}{p}}\right)\|\boldsymbol{T}^\star\|_{\mathrm{F}} \tag{11a}$$

$$\left\|\boldsymbol{T}^t - \boldsymbol{T}^\star\right\|_{\infty} \lesssim \left(C_3\rho^t + C_4\frac{\sigma}{\lambda_{\min}^\star}\sqrt{\frac{d\log d}{p}}\right)\|\boldsymbol{T}^\star\|_{\infty} \tag{11b}$$

*hold simultaneously for all $0 \le t \le t_0 = d^5$. Here, $0 < C_1, C_3, \rho < 1$ and $C_2, C_4 > 0$ are some absolute constants.*

We shall take a moment to discuss the merits of our approach in comparison to prior work. One of the best-known polynomial-time algorithms is the degree-6 level of the sum-of-squares hierarchy, which seems to match the computationally feasible limit in terms of the sample complexity [4]. However, this approach has a well-documented limitation in that it involves solving a semidefinite program of dimensions $d^3 \times d^3$, which requires enormous storage and computation power. Yuan et al. [73, 74] proposed to consider tensor nuclear norm minimization, which provably allows for reduced sample complexity. The issue, however, is that computing the tensor nuclear norm itself is already computationally intractable. The work [50] alleviates this computational burden by resorting to a clever unfolding-based spectral algorithm; it is a nearly linear-time procedure that enables near-minimal sample complexity (among polynomial-time algorithms), although it does not achieve exact recovery even in the absence of noise. The two-stage algorithm developed by [68] — which is based on spectral initialization followed by tensor power methods — shares similar advantages and drawbacks as [50]. The work [36] used tensor power methods for initialization, which, however, requires a large number of restart attempts; see discussions in [7]. Further, [67] proposes a polynomial-time nonconvex algorithm based on gradient descent over Grassmann manifold (with a properly regularized objective function), which is an extension of the nonconvex matrix completion algorithm proposed by [40, 41] to tensor data. The theory provided in [67], however, does not provide explicit computational complexities. The recent work [59] attempts tensor estimation via a collaborative filtering approach, which, however, does not enable exact recovery even in the absence of noise.

## 1.5 Notations

Before proceeding, we gather a few notations that will be used throughout this paper. For any tensors $\boldsymbol{T}, \boldsymbol{R} \in \mathbb{R}^{d \times d \times d}$, the inner product is defined as $\langle \boldsymbol{T}, \boldsymbol{R} \rangle = \sum_{j,k,l} T_{j,k,l} R_{j,k,l}$. The Frobenius norm of $\boldsymbol{T}$ is defined as $\|\boldsymbol{T}\|_{\mathrm{F}} := \sqrt{\langle \boldsymbol{T}, \boldsymbol{T} \rangle}$. For any vectors $\boldsymbol{u}, \boldsymbol{v} \in \mathbb{R}^d$, we define the vector products of a tensor $\boldsymbol{T} \in \mathbb{R}^{d \times d \times d}$ — denoted by $\boldsymbol{T} \times_3 \boldsymbol{u} \in \mathbb{R}^{d \times d}$ and $\boldsymbol{T} \times_1 \boldsymbol{u} \times_2 \boldsymbol{v} \in \mathbb{R}^d$ — such that

$$\left[\boldsymbol{T} \times_3 \boldsymbol{u}\right]_{ij} := \sum_{1 \le k \le d} T_{i,j,k} u_k, \qquad 1 \le i, j \le d; \tag{12a}$$

$$\left[\boldsymbol{T} \times_1 \boldsymbol{u} \times_2 \boldsymbol{v}\right]_k := \sum_{1 \le i,j \le d} T_{i,j,k} u_i v_j, \qquad 1 \le k \le d. \tag{12b}$$

For any $\boldsymbol{U} = [\boldsymbol{u}_1, \cdots, \boldsymbol{u}_r] \in \mathbb{R}^{d \times r}$ and $\boldsymbol{V} = [\boldsymbol{v}_1, \cdots, \boldsymbol{v}_r] \in \mathbb{R}^{d \times r}$, we define

$$\boldsymbol{T} \times_1^{\mathsf{seq}} \boldsymbol{U} \times_2^{\mathsf{seq}} \boldsymbol{V} := [\boldsymbol{T} \times_1 \boldsymbol{u}_i \times_2 \boldsymbol{v}_i]_{1 \le i \le r} \in \mathbb{R}^{d \times r}. \tag{13}$$

Further, $f(n) \lesssim g(n)$ or $f(n) = O(g(n))$ means that $|f(n)/g(n)| \leq C_1$ for some constant $C_1 > 0$; $f(n) \gtrsim g(n)$ means that $|f(n)/g(n)| \geq C_2$ for some constant $C_2 > 0$; $f(n) \asymp g(n)$ means that $C_1 \leq |f(n)/g(n)| \leq C_2$ for some constants $C_1, C_2 > 0$.

## 2 Initialization

This section presents formal details of the proposed two-step initialization. Recall that the proposed initialization procedures consist of two steps, which we detail separately.

### 2.1 Step 1: subspace estimation via a spectral method

The spectral algorithm is often applied in conjunction with simple "unfolding" (or "matricization") to estimate the *subspace* spanned by the $r$ factors $\{\boldsymbol{u}_i^\star\}_{1 \leq i \leq r}$. This strategy is partly motivated by prior approaches developed for covariance estimation with missing data [48, 50], and has been investigated in detail in [6]. For self-containedness, we provide a brief introduction below, and refer the interested reader to [6] for in-depth discussions.

Let

$$\boldsymbol{A} = \mathsf{unfold}^{1 \times 2}\left(\tfrac{1}{p}\boldsymbol{T}\right) \in \mathbb{R}^{d \times d^2}, \quad \text{or more concisely} \quad \boldsymbol{A} = \mathsf{unfold}\left(\tfrac{1}{p}\boldsymbol{T}\right) \in \mathbb{R}^{d \times d^2} \tag{14}$$

be the mode-1 matricization of $p^{-1}\boldsymbol{T}$ (namely, $\frac{1}{p}T_{i,j,k} = A_{i,(j-1)d+k}$ for any $1 \leq i, j, k \leq d$) [43]. The rationale of this step is that: under our model, the unfolded matrix $\boldsymbol{A}$ obeys

$$\mathbb{E}[\boldsymbol{A}] = \mathsf{unfold}\left(\boldsymbol{T}^\star\right) = \sum_{i=1}^{r} \boldsymbol{u}_i^\star \left(\boldsymbol{u}_i^\star \otimes \boldsymbol{u}_i^\star\right)^\top =: \boldsymbol{A}^\star, \tag{15}$$

whose column space is precisely the span of $\{\boldsymbol{u}^\star\}_{1 \leq i \leq r}$. This motivates one to estimate the $r$-dimensional column space of $\mathbb{E}[\boldsymbol{A}]$ from $\boldsymbol{A}$. Towards this, a natural strategy is to look at the principal subspace of $\boldsymbol{A}\boldsymbol{A}^\top$. However, the diagonal entries of $\boldsymbol{A}\boldsymbol{A}^\top$ bear too much influence on the principal directions and need to be properly down-weighed. The current paper chooses to work with the principal subspace of the following matrix that zeros out all diagonal components:

$$\boldsymbol{B} := \mathcal{P}_{\mathsf{off\text{-}diag}}(\boldsymbol{A}\boldsymbol{A}^\top), \tag{16}$$

where $\mathcal{P}_{\mathsf{off\text{-}diag}}(\boldsymbol{Z})$ extracts out the off-diagonal entries of a squared matrix $\boldsymbol{Z}$. If we let $\boldsymbol{U} \in \mathbb{R}^{d \times r}$ be an orthonormal matrix whose columns are the top-$r$ eigenvectors of $\boldsymbol{B}$, then $\boldsymbol{U}$ serves as our subspace estimate. See Algorithm 2 for a summary of the procedure, and [6] for in-depth discussions.

### 2.2 Step 2: retrieval of low-rank tensor factors from the subspace estimate

#### 2.2.1 Procedure

As it turns out, it is possible to obtain rough (but reasonable) estimates of all low-rank tensor factors $\{\boldsymbol{u}_i^\star\}_{1 \leq i \leq r}$ — up to global permutation — given a reliable subspace estimate $\boldsymbol{U}$. This is in stark contrast to the low-rank matrix recovery case, where there exists some global rotational ambiguity that prevents us from disentangling the $r$ factors of interest.

We begin by describing how to retrieve *one* tensor factor from the subspace estimate — a procedure summarized in RETRIEVE-ONE-TENSOR-FACTOR(). Let us generate a random vector from the provided subspace $\boldsymbol{U}$ (which has orthonormal columns), that is,

$$\boldsymbol{\theta} = \boldsymbol{U}\boldsymbol{U}^\top \boldsymbol{g}, \qquad \boldsymbol{g} \sim \mathcal{N}(\boldsymbol{0}, \boldsymbol{I}_d). \tag{17}$$

The rescaled tensor data $p^{-1}\boldsymbol{T}$ is then transformed into a matrix via proper "projection" along this random direction $\boldsymbol{\theta}$, namely,

$$\boldsymbol{M} = \tfrac{1}{p}\boldsymbol{T} \times_3 \boldsymbol{\theta} \in \mathbb{R}^{d \times d}. \tag{18}$$

Our estimate for a tensor factor is then given by $\lambda^{1/3}\boldsymbol{\nu}$, where $\boldsymbol{\nu}$ is the leading singular vector of $\boldsymbol{M}$ obeying $\langle \boldsymbol{T}, \boldsymbol{\nu}^{\otimes 3} \rangle \geq 0$, and $\lambda$ is taken as $\lambda = \langle p^{-1}\boldsymbol{T}, \boldsymbol{\nu}^{\otimes 3} \rangle$. Informally, $\boldsymbol{\nu}$ reflects the direction of the component $\boldsymbol{u}_i^\star$ that exhibits the largest correlation with the random direction $\boldsymbol{\theta}$, and $\lambda$ forms an estimate of the corresponding size $\|\boldsymbol{u}_i^\star\|_2$. We shall provide intuition in the full version [7].

A challenge remains, however, as there are oftentimes more than one tensor factors to estimate. To address this issue, we propose to re-run the aforementioned procedure multiple times, so as to ensure that we get to retrieve each tensor factor of interest at least once. We will then apply a careful pruning procedure (i.e. PRUNE()) to remove redundancy.

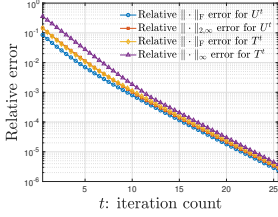
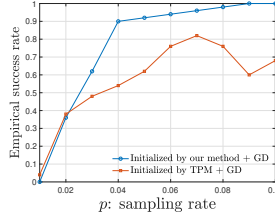
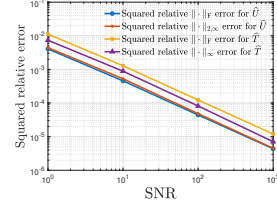

Figure 1: Relative error of the estimate $U^t$ and $T^t$ vs. the iteration count $t$ for the noiseless case, where $d = 100, r = 4, p = 0.1$.

Figure 2: Empirical success rate vs. sampling rate. Each point is averaged over 100 trials.

Figure 3: Squared relative errors vs. SNR for noisy settings. Here, $d = 100, r = 4, p = 0.1$. Each point is averaged over 100 trials.

## 3 Numerical experiments

We carry out a series of numerical experiments to corroborate our theoretical findings. We generate the truth $T^\star = \sum_{1 \le i \le r} u_i^{\star \otimes 3}$ randomly with $u_i^\star \overset{\text{i.i.d.}}{\sim} \mathcal{N}(0, I_d)$. The learning rates, the restart number and the pruning threshold are taken to be $\eta_t \equiv 0.2, L = 64, \epsilon_{\text{th}} = 0.4$.

We start with numerical convergence rates of our algorithm in the absence of noise. Set $d = 100$, $r = 4$ and $p = 0.1$. Fig. 1 shows the numerical estimation errors vs. iteration count $t$ in a typical Monte Carlo trial. Here, 4 kinds of estimation errors are reported: (1) the relative Euclidean error $\frac{\text{dist}_{\text{F}}(U^t, U^\star)}{\|U^\star\|_{\text{F}}}$; (2) the relative $\|\cdot\|_{2,\infty}$ error $\frac{\text{dist}_{2,\infty}(U^t, U^\star)}{\|U^\star\|_{2,\infty}}$; (3) the relative Frobenius norm error $\frac{\|T^t - T^\star\|_{\text{F}}}{\|T^\star\|_{\text{F}}}$; (4) the relative $\ell_\infty$ error $\frac{\|T^t - T^\star\|_\infty}{\|T^\star\|_\infty}$. Here, we set $T^t = \sum_{i=1}^r u_i^t \otimes u_i^t \otimes u_i^t$ with $U^t = [u_1^t, \cdots, u_r^t]$. For all these metrics, the numerical estimation errors decay geometrically fast.

Next, we study the phase transition (in terms of the success rates for exact recovery) in the noise-free settings. For the sake of comparisons, we also report the numerical performance of tensor power method (TPM) followed by gradient descent. When running the tensor power method, we set the iteration number and restart number to be 16 and 64 respectively. Set $r = 4$. Each trial is claimed to succeed if the relative $\ell_2$ error $\frac{\text{dist}_{\text{F}}(\widehat{U}, U^\star)}{\|U^\star\|_{\text{F}}} \le 0.01$. Fig. 2 plots the empirical success rates over 100 independent trials. As can be seen, our initialization algorithm outperforms the tensor power method.

The third series of experiments concerns the statistical accuracy of our algorithm. Take $t_0 = 100$, $d = 100, r = 4$ and $p = 0.1$. Define the signal-to-noise ratio (SNR) to be $\text{SNR} = \frac{\|T^\star\|_{\text{F}}^2 / d^3}{\sigma^2}$. We report in Fig. 3 three types of squared relative errors (namely, $\frac{\text{dist}_{\text{F}}^2(\widehat{U}, U^\star)}{\|U^\star\|_{\text{F}}^2}$, $\frac{\text{dist}_{2,\infty}^2(\widehat{U}, U^\star)}{\|U^\star\|_{2,\infty}^2}$ and $\frac{\|\widehat{T} - T^\star\|_\infty^2}{\|T^\star\|_\infty^2}$) vs. SNR. Here, the SNR varies from 1 to 1000. Figure 3 illustrates that all three types of relative squared errors scale inversely proportional to the SNR, which is consistent with our theory.

## 4 Discussion

The current paper uncovers the possibility of efficiently and stably completing a low-CP-rank tensor from partial and noisy entries. Perhaps somewhat unexpectedly, despite the high degree of nonconvexity, this problem can be solved to optimal statistical accuracy within nearly linear time. To the best of our knowledge, this intriguing message has not been shown in the prior literature. The insights and analysis techniques developed in this paper might also have implications for other nonconvex algorithms [36, 66, 69, 54, 67, 38, 61, 71, 37, 70] and other tensor recovery problems [2, 3, 63, 33, 60, 26, 75, 42, 77, 31, 10, 30].

## Acknowledgements

Y. Chen is supported in part by the AFOSR YIP award FA9550-19-1-0030, by the ONR grant N00014-19-1-2120, by the ARO grant W911NF-18-1-0303, by the NSF grants CCF-1907661 and IIS-1900140. H. V. Poor is supported in part by the NSF grant DMS-1736417.

## Footnotes

[1] We focus on symmetric order-3 tensors primarily for simplicity of presentation. Many of our findings naturally extend to the more general case with asymmetric tensors of possibly higher order.

[2] Here, a tensor $\boldsymbol{T} \in \mathbb{R}^{d \times d \times d}$ is said to be symmetric if $T_{j,k,l} = T_{k,j,l} = T_{k,l,j} = T_{l,k,j} = T_{j,l,k} = T_{l,j,k}$.

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
