[Reviews · NeurIPS 2019]

Reviewer 1



This paper presents an interesting finding that is backed up by rigorous analysis and reasonably persuasive examples. Although the idea is quite simple (vanilla gradient descent with warm enough initialization), the execution is precise and the writing quality is very high. I find the theoretical contributions to be clear and elegant, while the data analysis is a bit lacking. In particular, the lack of convincing real data analysis makes it unclear how how realistic the assumptions are/how strong the theoretical statements are in practice. However, the assumptions are rather standard, and it is unsurprising that requiring many random restarts etc are necessary for difficult non-convex settings. Finally, it is not totally clear exactly which results extend to non-symmetric tensors, though I assume many of the choices in this regard are for exposition.

Reviewer 2



In this paper, a fast nonconvex algorithm along with theoretical guarantees on local convergence and linear time computational complexity are developed and analyzed for symmetric tensor completion. The performance of the proposed algorithm is evaluated by conducting numerical tests on synthetic data and it is shown that the proposed method has some merits for dealing with noisy data. Overall, the paper is well-written and easy to read. The motivations are clear and the technical content of the paper seems to be correct. However, there are some issues that need to be addressed to further improve the current paper. - The experimental results are very limited. The main drawback of this paper is that there is no comparison with existing methods, which limits their contribution. - It is unclear whether the proposed method is really useful in practice. More experiments need to be done on real data. - Except for the theoretical analysis, it would be better to show the efficiency of the proposed algorithm with experimental results. - There is a discrepancy between the title of the paper that promises a tensor completion method and the proposed algorithm that is done only for symmetric tensor, and hence completely neglects the difference of decompositions to deal with symmetric tensor.

Reviewer 3



This paper studies the low-rank tensor completion problem. Authors propose a two-stage nonconvex algorithm, where the first stage is initialization and the second stage is gradient descent. Authors prove that sample complexity is same as existing works, but this algorithm enjoys linear convergence. Overall, the paper is readable and has good quality, but I suspect whether it is suitable for conference paper. One reason is that many key intuitions and backgrounds, especially for initialization step, are deferred to supplementary material. So the story is incomplete and has no clear explanations. Besides this, I have some technical concerns. 1, the most pressing concern is assumption A3. Can we relax A3 since that's too strong. It is not usually required in other related methods. Is it because of A3 that you don't need regularization in GD? Under A3, the least square-like problem has no design. 2, Q1 and Q2 in Sec 1.2 are not well proposed. It would be better if authors clarify contributions more clearly. To me, the main challenge is sample complexity, which is not improved. It's not surprising that two-stage algorithm achieves linear rate with certain near-optimal statistical rate, since in principle linear rate comes from applying GD on square objective. This nonconvex objective appears in low-rank matrix completion and robust estimation quite often, and has been studied in [33]. 3, there is no clear dependence with kappa in main theorem 1.4. Please clarify how the result depends on kappa. What's the power of kappa in step size. 4, why GD has no projection step such that the incoherence condition keeps holding along the iteration. Also, why we have an upper bound for the iteration times? This is not common for two-stage algorithms when it is applied to other problems, e.g. robust estimation. 5, A separate result for initialization step is needed. Using this initialization for other methods, e.g. [33], will the performance of other methods get improved? 6, for experiments, authors should provide comparisons with other advanced methods, especially [33, 63], to show the superiority.

[Author Response · NeurIPS 2019]

We thank the reviewers for very helpful comments. This letter addresses several major questions raised by the reviewers.

**Symmetric tensor models.** We will change our title to "symmetric tensor completion ..." as suggested. Note, however,
that our results can be extended to the non-symmetric cases straightforwardly. We shall include a new section to discuss
this generalization in the final paper. In addition, it has been shown by [4] that any algorithm for symmetric tensors can
be used to estimate asymmetric tensors.

**Comparisons with prior algorithms.** We will add detailed numerical comparisons with prior algorithms to show the
advantage of our algorithm (numerical comparison with tensor power method (TPM) [33] is shown in the figure below).
The theoretical comparison when $r = O(1)$ (ignoring log factors) is summarized in the table below. Our algorithm
achieves optimality in all three aspects (computational cost, sample complexity, statistical accuracy).

| | sample complexity | computational cost | $\ell_2$ error (noisy case) | |
|---|---|---|---|---|
| our theory | $d^{1.5}$ (computational limit) | $pd^3$ (linear time) | $\sigma\sqrt{\frac{d}{p}}$ (optimal) |  |
| [33] (TPM + alt-min) | $\frac{d^{1.5} \text{ (initialized by } our\ scheme)}{d^3 \text{ (initialized by TPM, detailed below)}}$ | $pd^3$ | n/a | |
| [4], [50] (SOS) | $d^{1.5}$ | $d^{10}$ | $\sigma d^{1.5}$ | |
| [63] (spectral init + GD on manifold) | $d^{1.5}$ | polynomial ($\gg$ linear time) | n/a | |
| [48] (convex relaxation + unfolding) | $d^2$ | $d^5$ | n/a | |

**Real-data experiments.** We will adopt the reviewers' suggestion to include experimental results on real data, including
two applications. 1. *Medical images*: We will apply our algorithm to MRI scans to exploit similarities between different
slices of MRI. We plan to use the MRI scans from the OsiriX repository as our dataset. We will first stack images into a
tensor and then apply our algorithm to complete the tensor. The recovered images as well as the relative square error
will be presented. 2. *Collaborative filtering*: We will apply our algorithm to collaborative filtering in the tensor case,
where each user corresponds to two categories. For example, we can use our algorithm to predict customer ratings for
hotels as well as attributes they care (e.g. room/location/service).

**Inadequecy of the tensor power method (TPM) for initialization.** In order for GD or other first-order methods to
converge fast, it is crucial to provide a careful initialization (as we show in the supplement, random initialized GD does
not work optimally). One approach that has been adopted in prior work is TPM. However, as already pointed out by
[53], the TPM performs quite suboptimally for tensor problems (even when there is no missing data). As discussed in
Section 4.2 of the supplementary material, when using the TPM in the initialization step (e.g. [33]), the perturbation
bound in [2, Theorem 5.1] requires the tensor perturbation to be no larger than $o(1/d)$. This cannot possibly hold under
the sample size assumption $p \asymp \text{poly} \log d / d^{1.5}$ (since we can only hope to get $1/\text{poly} \log d$ even in the noiseless case
[33, Theorem 2.1]). As a result, one would need an unaffordable large number of random restarts if we were to adopt
the approach in [33] initialized by the TPM. Instead, the approach in [33] can work as long as we replace the TPM
initialization by the initialization procedure proposed in our work (numerical comparison is shown in the figure above).

**Specific questions by Reviewer 1**: 1. *Initialization*: see the response above about "initialization".

2. *Number of restarts*: We use 10 restarts throughout all experiments, which suffice to obtain satisfactory performances.
As predicted by the theoretical analysis, a constant number of restarts are sufficient. This should be contrasted with
prior work which requires a huge number of restarts (see discussion about "initialization" above).

**Specific questions by Reviewer 2**: see the response above for "real-data experiments".

**Specific questions by Reviewer 3**: 1. *Dependency on the condition number $\kappa$*: The quantity $\kappa$ only affects the sample
size requirement. If we remove the assumption A3, then an extra $\text{poly}(\kappa)$ factor will appear in the sample complexity
(more specifically, $p \gg \kappa^6 d^{-1.5} \log^6 d$) in Theorem 1.4. In addition, we note that $\kappa$ does not affect the step size.

2. *Initialization*: see the response above for "initialization". We will separate our theoretical results for initialization
and local optimization to improve readability.

3. *Contribution*: The sample complexity $O(d^{1.5})$ is widely conjectured to be optimal among all polynomial algorithms
when $r = O(1)$. However, it is unclear how to achieve it within linear time. In particular, many nonconvex algorithms
require an initialization stage that is very expensive both in sample complexity and computational complexity (see
discussion above about the TPM and theoretical comparisons with other algorithms). Our main contribution is to
develop a provably efficient linear-time algorithm with minimal sample complexity.

4. *Implicit regularization of GD*: It has been shown that there is no need to enforce projection or other regularization in
various statistical models (e.g. matrix completion and phase retrieval) [46]. Similarly, a projection step is not crucial for
the tensor case (which can be proven via a leave-one-out analysis). Regarding the upper bound on the iteration count:
this is mainly due to the presence of noise. In the noiseless case, there is no need to impose any upper bound on $T$; in
the noisy case, one can also use a slightly more complicated argument to remove this upper bound (we have chosen to
keep this upper bound so as to slightly simplify the analysis, as $T = O(d^{10})$ seems to cover all practical scenarios).

[Meta-Review · NeurIPS 2019]

This paper gives a new algorithm for low rank tensor completion. Compared to previous works it is faster and achieves tight sample complexity. Solid contribution.